# A Comparison of Two Different FFPE Tissue Dissection Methods for Routine Diagnostics in Molecular Pathology: Manual Macrodissection versus Automated Microdissection Using the Roche “AVENIO Millisect” System

**DOI:** 10.3390/cancers15123249

**Published:** 2023-06-20

**Authors:** Jan Jeroch, Tobias Riedlinger, Christina Schmitt, Silvana Ebner, Ria Winkelmann, Peter J. Wild, Melanie Demes

**Affiliations:** 1Wildlab, University Hospital Frankfurt MVZ GmbH, Theodor-Stern-Kai 7, 60596 Frankfurt am Main, Germany; silvana.ebner@kgu.de (S.E.); peter.wild@kgu.de (P.J.W.); melanie-christin.demes@kgu.de (M.D.); 2Dr. Senckenberg Institute of Pathology, University Hospital Frankfurt, Theodor-Stern-Kai 7, 60596 Frankfurt am Main, Germany; t.riedlinger@hotmail.com (T.R.); christinastefanie.schmitt@kgu.de (C.S.); ria.winkelmann@kgu.de (R.W.)

**Keywords:** FFPE tissue, tissue dissection, manual macrodissection, automated microdissection

## Abstract

**Simple Summary:**

In order to facilitate routine processes not only by saving time and personnel capacities but also by minimising the loss of valuable patient material, automated systems have become increasingly attractive for integration into laboratory workflows. The aim of this study is to assess whether the automatic dissection with the help of the “AVENIO Millisect” system has advantages over the manual dissection data of the same samples, and directly compare both processes.

**Abstract:**

Currently, in routine diagnostics, most molecular testing is performed on formalin-fixed, paraffin-embedded tissue after a histomorphological assessment. In order to find the best possible and targeted individual therapy, knowing the mutational status of the tumour is crucial. The “AVENIO Millisect” system Roche introduced an automation solution for the dissection of tissue on slides. This technology allows the precise and fully automated dissection of the tumour area without wasting limited and valuable patient material. In this study, the digitally guided microdissection was directly compared to the manual macrodissection regarding the precision and duration of the procedure, their DNA concentrations as well as DNA qualities, and the overall costs in 24 FFPE samples. In 21 of 24 cases (87.5%), the DNA yields of the manually dissected samples were higher in comparison to the automatically dissected samples. Shorter execution times and lower costs were also benefits of the manual scraping process. Nevertheless, the DNA quality achieved with both methods was comparable, which is essential for further molecular testing. Therefore, it could be used as an additional tool for precise tumour enrichment.

## 1. Introduction

In order to select a helpful targeted and individual cancer therapy, it is essential to know the mutation status of cancer-related genes [1,2]. Today, most molecular genetic testing is performed with material extracted from formalin-fixed, paraffin-embedded (FFPE) tissue after a histological assessment due to its conserving properties [3,4,5]. However, with formalin leading to DNA modifications as well as to low DNA quantities and fragmentation, FFPE material is often limited to certain methods only and artifacts have to be kept in mind [3,4,6,7,8,9]. Assays, nevertheless, still rely on FFPE tissue, since there often is not a possibility to obtain fresh material [3,4,10,11]. The most frequent used fast and low-priced tumour enrichment method is manual tissue dissection by directly scraping off FFPE tissue from glass object slides with the help of a scalpel [1,12,13,14]. However, cases with lower tumour content or where the tumour cannot be dissected by hand require a more precise approach [12,15]. To improve the sensitivity of molecular analyses and reduce the need for additional biopsies, the neoplastic cellularity in the sample should be maximised, making it highly important that tumour cells are isolated adequately from its heterogeneous, more dominant benign tissue microenvironment [1,16,17]. Tumour heterogeneity is extremely common and has a consequence for treatment responses and prognoses [18,19]. Here, an alternative such as the laser capture microdissection (LCM) could be applied [12,13]. LCM enables the isolation of a single cell subgroup or even a single cell in a short period of time without morphologically altering them and reduces the danger of tissue loss [19,20,21]. Yet, the LCM is still not widely adopted due to the high costs and labour-intensiveness it entails [1,12].

In 2017, Roche (Basel, Switzerland) released the new automated “AVENIO Millisect System”. This automated high-performance system was introduced as a CE-IVD approved method to dissect both paraffinised and deparaffinised tissues [5,12]. With an object table with space for up to four object slides, one of which is hematoxylin and eosin-stained (H&E), three different milling tip sizes, an integrated camera with zooming function and an automatic report generation, the system provides an easy, flexible and simultaneously accurate dissection process with a precision of 250 µm^2^. The manufacturer also states the dissection time for one object slide to be approximately 2 min, which is, however, dependent on the tumour area.

In order to facilitate routine processes not only by saving time and personnel capacities but also by minimising the loss of valuable patient material, automated systems have become increasingly attractive for integration into laboratory workflows. A study by Geiersbach et al. aimed to compare traditional manual macrodissection with digitally guided microdissection on a series of FFPE pancreatic adenocarcinomas and highlighted the much higher tumour enrichment after the digitally guided approach [12]. Qi et al. compared the “AVENIO Millisect” instrument with the manual dissection on breast cancer samples and concluded that both are comparable concerning DNA yield and quality [5]. Krizman et al. contrasted the laser-based dissection, the automated and the manual dissection from lung cancer blocks [1]. They demonstrated the highest dissection resolution for the laser procedure. Manual scraping did not provide pure tumour cells and therefore had the lowest resolution. The aim of this study was to assess whether the automatic dissection with the help of the “AVENIO Millisect” system has advantages over the manual dissection data of the same samples, and directly compare both processes.

## 2. Materials and Methods

### 2.1. Case Selection and Slide Preparation

The tissues randomly used in this study were approved by the institutional Review Boards of the UCT and the Ethical Committee at the University Hospital Frankfurt and routinely formalin-fixed and paraffin-embedded for immunohistochemical, histomorphological and genetic analyses. After sectioning these blocks using a microtome, the slices with a thickness of 3 µm were transferred onto glass object slides and H&E-stained, the tumour contents were confirmed and approved microscopically by pathologists and the respective areas were marked. The collective of investigated samples contained 24 different FFPE blocks of 24 different patients and 10 different tumour entities as described in the Results section.

### 2.2. Deparaffinisation and Dissection

Before the tissue was dissected manually and automatically from the slides, it was deparaffinised. Therefore, the slides were incubated at 70 °C for 30 min, followed by an incubation of 20 min in xylene (Thermo Fisher Scientific, Waltham, MA, USA), 10 min in isopropanol (Sigma-Aldrich, St. Louis, MO, USA) and 10 min of air-drying at room temperature each. Afterwards, the same amount of slides was dissected per sample with a scalpel as well as with the “AVENIO Millisect” System (Roche, Basel, Switzerland). In order to carry this out, pathologists previously identified the specific tumorous regions with the help of a microscope. For the manual microdissection, FFPE tissue was scraped into Eppendorf tubes for subsequent proteinase K digestion. After setting the parameters for the automated dissection, such as the thickness of the respective sections and whether the tissue is paraffinised or not, 1–3 slides of the same block and one H&E-stained slide were placed in the machine. With the help of an included camera, images of both stained and unstained slides were taken. The software (Version 2.0.0) enabled an easy alignment of the stained reference image with the unstained slides. Afterwards, the tumorous regions were digitally marked for dissection. The software calculated the expected dissection area, time and the amount of dissection fluid (Roche, Basel, Switzerland), which, along with the right-sized milling tip and a collection tube, needed to be provided. The milling tip, which was available in three different sizes, was moved to the filling station, where dissection fluid was sucked up. While cutting the marked areas, the material was aspirated simultaneously into the tip containing the dissection fluid and the whole content was transferred to the collection tube. A centrifugation step was performed to get rid of the supernatant dissection fluid. During the dissection, the process was displayed live. After the whole procedure, another set of images was taken, which enabled the direct comparison of the tissue areas before and after the dissection process. In the end, a report was generated for each run and sample, respectively, including further parameters such as the single area sizes, tip size and aspiration speed.

### 2.3. DNA Extraction, Quantification and Qualification

In the next step, the tissue of both dissection methods was digested with proteinase K and the nucleic acid was extracted with the help of the MaxWell^®^ RSC instrument (Promega, Madison, WI, USA) and the Maxwell^®^ FFPE Plus LEV DNA Purification Kit (Promega, Madison, WI, USA) according to the manufacturer’s instructions. The concentration of the extracted DNA was determined using the Qubit 2.0 Fluorometer (Thermo Fisher Scientific, Waltham, MA, USA) and their quality via fragment analysis (ABI 3130xl Genetic Analyzer (Thermo Fisher Scientific, Waltham, MA, USA)) according to their respective protocols.

## 3. Results

In order to enable a direct comparison of the two dissection methods, a random collective of 24 different FFPE samples of both sexes (female = 14, male = 10) and 11 different tumour entities was dissected both automatically and manually each. The majority of cases were colonic and colorectal adenocarcinomas, as well as malignant melanomas (four each), followed by non-small cell lung cancers (NSCLC) and thyroid carcinomas (three each), ovarian carcinoma (two), as well as breast cancer, endometrioid adenocarcinoma, pancreatic and prostatic adenocarcinoma (one each), reflecting daily routine diagnostics (Table 1). The age span ranged between 28 and 90 years with a mean of 57.2 and a median of 56 years.

Table 2 compares the DNA concentrations of the 24 extracted samples between the two dissection methods and additionally lists the dissected area and the dissection time of the automated procedure. The mean DNA concentration yielded 19.37 ng/µL (median 10.2 ng/µL), with a minimum of 0.76 ng/µL and a maximum of 120 ng/µL, for the automatically dissected and 53.82 ng/µL (median 37.25 ng/µL), with a minimum of 0.484 ng/µL and a maximum of 216 ng/µL, for the manually dissected tissues. In 87.5% (21/24 samples), the manually scraped tissue yielded higher DNA concentrations. Samples 18, 19 and 23 were the only ones where the DNA concentration of the manually dissected tissue was smaller than for the automatically dissected (cf. also Figure 1):

The mean fragment length of the DNA as determined using the ABI 3130xl Genetic Analyzer was 250 bp and therefore showed a sufficient quality for further analyses. The total areas range between a minimum of 53 mm^2^ and a maximum of 473 mm^2^ with a mean of 201 mm^2^. The automatic dissection time varied between 4 min and 27 s and 17 min and 4 s for one sample and averaged 08:20 min, depending on the area size.

Figure 2 shows the workflow of the exemplary sample 15 during the automated dissection process. After the position on stage was marked with a slide marking pen (Figure 2, second from the left) using the H&E-stained slide as a reference (Figure 2, left), the dissection path was pre-set for the instrument with the included software (Figure 2, second from the right) in order to facilitate the dissection procedure. Afterwards, another image was taken for the visualisation of the dissection precision (Figure 2, right). The dissected tissue volume for this sample was 0.869 mm^3^ and the respective total tissue area 289.6 mm^2^.

## 4. Discussion

The “AVENIO Millisect” System Roche provides a reliable helping tool for the precise tissue dissection of target areas and thus supports the attainment of clinically relevant information for molecular diagnostics without the need of further laboratory equipment. With its automatic and gentle workflow, the loss of valuable sample material is reduced, which is especially important for cases with small tumour regions. A dissection completeness and efficiency can be confirmed by the photographs taken before and after the dissection which is essential for quality assurance.

Opposed to what was expected, the automated dissection using the “AVENIO Millisect” instrument took significantly more time than the manual scraping process confirming the findings of Peng et al. With a mean of 08:20 min, it took even longer than the maximum of 7 min stated by the manufacturer. Depending on the size of the embedded material several object slides with tissue slices were needed, increasing the duration of the automatic dissection as well. Since the respective aspiration speed (time it took for the material to get aspirated into the milling tip) remained relatively constant, the automated dissection time was mainly dependent on the total amount of dissection areas and the number of slides per sample, as well as on the size of the tips. However, with the sum of areas ranging between a minimum of 53 mm^2^ and a maximum of 473 mm^2^ and a mean of 201 mm^2^, no clear coherence between total area and DNA concentration could be observed. Even though the area somehow corresponds to the number of cells, higher cell numbers do not automatically lead to higher DNA yields. Except for samples 18, 19 and 23, all other manually dissected samples resulted in higher concentrations than the automatically dissected ones. With 53.82 ng/µL, the mean DNA concentration of the manually dissected samples was more than two times higher than the mean concentration of the automatically dissected ones. With an average of 19.37 ng/µL, after the automatic dissection, the yield of DNA seemed to be sufficient for further molecular tests such as genetic analyses. However, using Qubit 2.0 for the quantification of DNA yields involved not only DNA fragments, but also unspecific other double-stranded fragments. Another reason could be the use of different buffers. The automated dissection needed a slightly higher amount of buffer, leading to a higher dilution and therefore lower concentrations of DNA.

The quality of DNA isolated from FFPE tissue is known to be limited since formalin modifies nucleic acids. However, assessing the quality of the extracted DNA obtained using the automatic dissection method via the 3130xl Genetic Analyzer proved that digitally guided microdissection does not further reduce their quality. While not all samples that are submitted for molecular testing are salvageable, some samples that could not be scratched manually might be successful following the digitally guided process. This is an important finding because the samples that would benefit the most from a more precise dissection technique are usually the ones limited in overall tumour content, leading to false negative results. Here, an efficient retrieval of nucleic acids is especially critical. Cancer patients may often have to undergo additional clinical surgeries in order to obtain a new adequate sample. Another big advantage of digitally guided procedures is their ability to provide a documentation of the whole process.

Consumables for the “AVENIO Millisect” device such as tips and the dissection fluid are specific and therefore more expensive than the single-use scalpels used for the manual dissection. For each slide, a new dissection tip is needed. However, an automated dissection with the routinely used ATL buffer (Qiagen N.V, Venlo, The Netherlands), which has not been tested outside this study yet, was also successful. However, many laboratories cannot afford any of the available electronic dissection instruments and have to rely on the manual procedure. Whether the costs of purchasing and operating such a device are acceptable mainly depends on the laboratory throughput, individual patient issues and clinical importance. Nevertheless, a precise high-resolution dissection up to single tumour cells is mostly not necessary for molecular testing.

Several studies have demonstrated the general imprecision and subjectivity of pathologists’ estimations of tumour contents, with overestimation errors being regarded as the most critical for potential false negative results in molecular testing [12,17,18]. The “AVENIO Millisect” system, just as the manual process, completely depends on these pathologists’ assessments.

## 5. Conclusions

FFPE tissue is still the most used sample material for molecular testing in routine diagnostics. However, the available patient material is often limited. In order to minimise the loss of this valuable material as well as time and personnel capacities, automated systems have become increasingly attractive for integration into laboratory workflows.

The “AVENIO Millisect” automation solution introduced by Roche allows the precise and fully automated dissection of the tumour area without unnecessarily wasting patient material.

Since this small cohort should reflect real cases of the broad spectrum of entities in routine diagnostics and both dissection methods require the same sample preparation via deparaffinisation and afterwards the same handling for proteinase k digestion and DNA extraction, the focus of this study was only on comparing the two different dissection methods. Together with the preparation of the instrument, the dissection using the automated system took significantly longer than the manual process. Sometimes, deparaffisisation may not be necessary for certain tests. In routine diagnostics, however, removing paraffin preferably completely is essential to improve the overall quality for subsequent complex analyses such as NGS.

Tumour heterogeneity and the irregular shape of the sample material is an issue appearing in every study and cannot be avoided. By analysing and comparing the same samples using the same H&E-stained reference, this problem can be reduced.

In the next step, a pair-wise statistical test would be appropriate to assess whether the difference in the DNA concentrations was statistically significant. Additionally, the measurement of the manually dissected tissue area with a respective scanner would be a good step to be able to compare the different areas and the corresponding DNA concentrations. The contamination of tumour DNA with benign DNA was not assessed and would be considered as a future step as well. As we did not decide to keep the instrument, we stopped at this point to avoid further costs.

Setting the costs and duration of one analysis with the “AVENIO Millisect” instrument aside, the system provides an easy and helpful tool for routine diagnostics in molecular pathology without impairing the quality of results. Whether this leads to a higher dissection resolution concerning quality and quantity and therefore increasing confidence in molecular results and an associated improved patient therapy in general could not be concluded yet and therefore depends on the case and clinical question. Nevertheless, the preconditions for a complex molecular analysis are fulfilled.

## Figures and Tables

**Figure 1 cancers-15-03249-f001:**
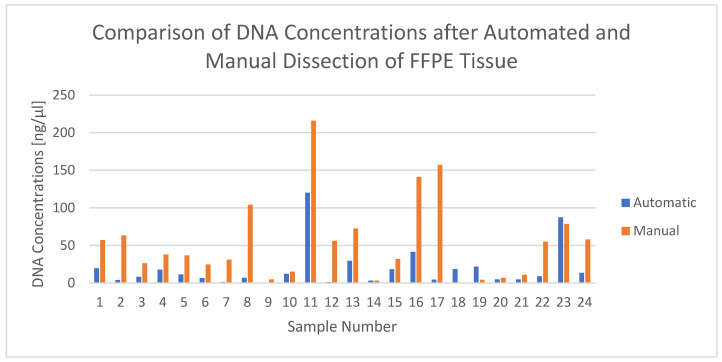
Comparison of DNA concentrations after automated and manual dissection of FFPE tissue. Concentrations in ng/µL are depicted on the *y*-axis, investigated sample numbers on the *x*-axis. Blue bars resemble the concentrations after the automatic dissection process, whereas orange bars display the concentrations of the manual procedure. Of all 24 samples, only samples 18, 19 and 23 showed a higher concentration after the automated dissection as compared to the manual approach.

**Figure 2 cancers-15-03249-f002:**
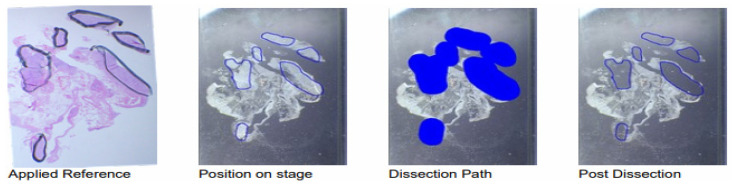
Images taken from sample 15 during the automated dissection process. Using the H&E-stained slide as reference (“Applied Reference”, (**left**)), the respective areas on the slides were marked with a slide marking pen (“Position on stage”, (**second from the left**)). With the help of the included software these previously identified regions were then denoted for the instrument (“Dissection Path”, (**second from the right**)). In this exemplary sample 15, the dissected area (blue) was 289.60 mm^2^. After the dissection procedure another image was taken in order to visualise dissection precision (“Post Dissection”, (**right**)), which was successful in this case.

**Table 1 cancers-15-03249-t001:** Collective of investigated FFPE samples for manual and automatic dissection. A total of 24 different samples of 11 different tumour entities was analysed.

Tumour Entity	Absolute Number of Cases (*n* = 24)
Breast Cancer	1
Colonic Adenocarcinoma	4
Colorectal Adenocarcinoma	4
Endometrioid Adenocarcinoma	1
Malignant Melanoma	4
Non-Small Cell Lung Cancer	3
Ovarian Carcinoma	2
Pancreatic Adenocarcinoma	1
Prostatic Adenocarcinoma	1
Thyroid Carcinoma	3

**Table 2 cancers-15-03249-t002:** DNA concentrations after automated and manual dissection, total area of dissection material and dissection time of automatic dissection process.

Sample ID	DNA Concentrations Automatically Dissected [ng/µL]	DNA Concentrations Manually Dissected [ng/µL]	Total Area Automatically Dissected [mm^2^]	Dissection Time Automatically Dissected [min]
1	19.5	57	211.60	11:03
2	3.83	63.2	110.40	05:54
3	8.05	26.3	68.90	04:55
4	17.7	38	107.80	05:44
5	11.4	36.5	53.00	04:29
6	6.44	24.6	337.20	07:31
7	1.34	30.9	120.20	06:13
8	6.76	104	238.30	05:53
9	0.76	4.7	195.90	11:14
10	12.2	15.1	160.40	10:44
11	120	216	413.10	11:43
12	1.14	56	262.20	13:28
13	29.4	72.5	321.30	11:27
14	3.05	3.13	55.20	04:27 (min)
15	18.2	31.8	289.60	09:47
16	41.5	141	473.00	14:02
17	4.33	157	203.90	08:13
18	18.6	0.484	241.80	06:33
19	21.8	4.14	291.20	17:04 (max)
20	4.58	6.82	91.90	05:27
21	4.68	10.9	90.90	05:28
22	9.04	55	232.00	06:17
23	87.3	78.7	159.80	07:09
24	13.4	58	93.30	05:23

DNA concentrations vary between a minimum of 0.76 ng/µL and a maximum of 120 ng/µL for the automatically dissected (mean: 19.37 ng/µL) and a minimum of 0.484 ng/µL and a maximum of 216 ng/µL (mean: 53.82 ng/µL) for the manually dissected samples. Dissection areas from the automated dissection range between 53 mm^2^ and 473 mm^2^. The shortest automatic dissection time is 4 min 27 s (sample 14) and the longest 17 min 4 s (sample 19).

## Data Availability

The data presented in this study are available in this article.

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
