# Peer review of "A Comparison of Two Different FFPE Tissue Dissection Methods for Routine Diagnostics in Molecular Pathology: Manual Macrodissection versus Automated Microdissection Using the Roche “AVENIO Millisect” System"

_cancers, 2023, doi:10.3390/cancers15123249_

Round 1
Reviewer 1 Report
I have two comments.
1. In the M&M section, no description of the manual macrodissection procedure was given. In particular, are "the specific previously identified tumourous regions" the same in manually vs. automatically dissected samples?
2. In 21 of 24 cases (87.5 %) the DNA yields of the manually dissected samples were higher in comparison to the automatically dissected samples. What is the cause for the difference? This needs to be addressed in the discussion
Author Response
Dear Reviewer,
thank you for your support and your valuable feedback. Please find attached my answers. Kind regards

Reviewer 2 Report
Comments:
An advantage of manual macro-dissection is that there is no need to deparaffinize the tissue sections. In this way the macro dissected tissue can be more easily handled. Moreover, if using an FFPE purification kit this should include the deparaffinisation step in the tube.
The sample selection is not ideal as it is too low per tumour. There are only a handful of samples for each tumour site which cannot be representative of the tumour of origin (eg colorectal cancer). Within each tumour type, tumours can vary significantly in terms of heterogeneity, morphology, infiltration by lymphocytes and contamination of benign or normal tissue. This represents a significant drawback.
Minor
1. Line 28-29 – Instead of stating that the AVENIO Millisect system did not “reduce” the DNA quality, it would be more accurate to state that the DNA qualities achieved with both methods was comparable.
2. Many in-text citations and references do not indicate the year of the publication. This information is critical when referencing.
3. Line 75 citation included twice.
4. Line 140 – 143: Here the median may offer a better representation of the data since it is less affected by outlier data. A maximum of 216ng/µL as opposed to 120ng/µL will inflate the mean.
5. The y-axis title of Figure 1 is not fully readable.
6. Line 184: The presented data and study does not provide evidence to support that the loss of sample material is reduced with the automated method. Were there different number of slides or tumour area used for each method?
7. For the benefit of the reader, it would be useful to include some more details about both the manual and the automated dissection methods. For macro-dissection: were the sections left to dry after the isopropanol? It is assumed that the scraped of tissue was then transferred to a tube? For the automated method: How does the machine provide the dissected specimen? Is this diluted in dissection fluid? Is there any preparation needed before proceeding with DNA extraction?
8. A pair-wise statistical test should be used to measure if the difference in the DNA concentration between the manual and the automated methods is significant.
9. Line 197-198: “no clear coherence between total area and DNA concentration could be observed”. Please indicate how this was assessed.
10. The slide order should be clearly outlined. Was each block sectioned to produce slides for one method then the other or were slides assigned to each method alternatively during sectioning? Consecutive sectioning results in changing tissue structures between the first slide and the last. Hence, if comparing to the first H&E, the last unstained slide may be expected to be quite different.
Major
1. An estimate of the manually dissected area is not reported. This metric is critical to normalize the tissue area inputted. If the manual procedure included more tissue area than the automated method, then DNA yield is expected to be higher and vice versa.
2. In such a study one would expect to include replicates from each tumour sample per method. Especially given the variations observed in the DNA concentration yields.
3. The study does not measure the contamination of tumour DNA with normal or benign DNA. The inclusion of a mutation quantification method of a previously identified mutation can provide insight on the accuracy of the methods.
4. Line 25: “their DNA concentrations as well as DNA qualities and the overall costs”. Very limited information is provided for costs. No values are provided to compare in the manuscript.
5. The evidence provided on the DNA quality is very limited (Line 153-154). With such limited data, on cannot state that the DNA quality is comparable between methods. More parameters for DNA quality should be reported.
Some sentences need revising as they are too long or missing punctuation.
Author Response

(The authors gave the same response as above.)

Reviewer 3 Report
Although I studied the manuscript with interest, in my opinion, such a study does not meet the goals and objectives of the journal. The study design also raises questions. Wouldn't it be better to take samples of one type of tumor of a certain organ? Different tumors have different numbers of cells and therefore are the source of different amounts of material.
Also, I did not find a conclusion section in the manuscript.
In general, the work has value but is suitable for another journal on a narrow, specific issue.
Author Response
Dear Reviewer,
thank you for your support and you valuable feedback. Please find attached my answers. Kind regards

Round 2
Reviewer 2 Report
Although some study limitations persist, the authors have adequately declared and discussed these. This description will further support the reader to provide information about the systems in place.
It is understood that the intention was to reflect the diagnostic setting, nonetheless a more robust study design would generate more informative and reliable results that can then adequately inform the diagnostic setting. This applies to inclusion of replicates, statistical analysis and other aspects.
Overall, the study is interesting and provides evidence around the use of an automated dissection system as opposed to a manual technique.
Reviewer 3 Report
As I have already reported, I read the manuscript with interest because the material is noteworthy and practically oriented. I had doubts about which I did not inform the authors but brought to the notice of the journal's editorial board.
Two representatives of the project (including the principal investigator) cooperate with the company whose product is evaluated in the work (Roche). With this in mind, I decided to reject the article on issues of academic integrity. My opinion remains the same on the second round of review.
I have no questions about the scientific content of the article.
Round 3
Reviewer 2 Report
Acceptable for publication
Reviewer 3 Report
I have no further comments. After the work on improving the manuscript and the explanations provided by the authors, the article can be accepted for publication.